# Atmospheric horizontal gradients measured with eight co-located GNSS stations and a microwave radiometer

Tong Ning[1] and Gunnar Elgered[2]

[1]Lantmäteriet (The Swedish Mapping, Cadastral and Land Registration Authority), SE-80182, Gävle, Sweden
[2]Department of Space, Earth and Environment, Chalmers University of Technology, Onsala Space Observatory, SE-43992 Onsala, Sweden.

**Correspondence:** T. Ning (tong.ning@lm.se)

**Abstract.**

We have used eight co-located GNSS stations, with different antenna mounts, to estimate atmospheric signal propagation delays in the zenith direction and linear horizontal gradients. The gradients are compared with the results from a water vapour radiometer (WVR). The water drops in the atmosphere have a negative influence on the retrieval accuracy of the WVR. Hence, we see better agreement using WVR data with a liquid water content (LWC) less than 0.05 mm compared to when LWC values of up to 0.7 mm are included. We have used two different constraints when estimating the linear gradients from the GNSS data. Using a weak constraint enhances the GNSS estimates to track large gradients of short duration at the cost of increased formal errors. To mitigate random noise in the GNSS gradients, we adopted a fusion approach averaging estimates from the GNSS stations. This resulted in significant improvements for the agreement with the WVR data, a maximum of 17 % increase in the correlation and an 14 % reduction in the root mean square (RMS) difference for the east gradients. The corresponding values for the north gradients are both 25 %. Overall, no large differences in terms of quality are observed for the eight GNSS stations. However, one station shows slightly poorer agreement for the north gradients compared to the others. This is attributed to the station's proximity to a radio telescope, which causes data loss of observations at low elevation angles in the south-south-west direction.

## 1 Introduction

After decades of development, data acquired from continuously operating global navigation satellite system (GNSS) stations are widely used in various applications, including precise positioning for navigation, real-time tracking for transportation logistics, environmental monitoring, including climate studies, geodesy, and geophysical research to understand the dynamics of the Earth's crust (Teunissen and Montenbruck, 2017). The equivalent total propagation delay in the zenith direction, referred to as the zenith total delay (ZTD), estimated from ground based GNSS observations in the Nordic countries are used within a state-of-the-art km-scale numerical weather prediction system (Lindskog et al., 2017). The investigation demonstrates that the assimilation of GNSS ZTD can benefit from enhancements in general data assimilation techniques. This can lead to improved forecast quality and more accurate numerical weather predictions (NWP). Furthermore, the usage of GNSS data to estimate horizontal tropospheric gradients has become increasingly prevalent in recent years. These gradients, which represent the asym-

metry of signal delays in the azimuth direction contain information on the local meteorological conditions, and are crucial for improving the accuracy of GNSS-derived ZTD. Studies have shown that the incorporation of GNSS tropospheric gradients can enhance meteorological applications, such as weather forecasting and climate research. For instance, Thundathil et al. (2024) found that the combined assimilation of horizontal gradients together with the ZTD into the weather research and forecasting model results in a clear improvement in the humidity field at altitudes above 2.5 km and therefore they recommended further GNSS gradient assimilation studies. With a relatively high temporal resolution, continuously improving spatial density, and less expensive receivers, ground-based GNSS networks are also used to monitor long-term changes in the atmospheric water vapour (Chen and Liu, 2016; Parracho et al., 2018; Jones et al., 2020). Furthermore, the remote sensing of the atmosphere at a given station improves with the addition of more GNSS constellations. This enhancement comes from increasing the number of simultaneous measurements and their distribution in various directions, which benefits gradient estimation. Ning and Elgered (2021) found that relative to the GPS-only solution, the solution using GPS, GLONASS, and Galileo data resulted in an increase in the correlation coefficient of 11 % for the east gradient and 20 % for the north gradient.

The Swedish GNSS network of continuously operating reference stations, SWEPOS$^{TM}$, was declared operational for post-processing applications and support for real-time positioning with metre accuracy in 1998. It is operated by Lantmäteriet (Swedish mapping, cadastral and land registration authority). Currently (February 2025) the SWEPOS network consists of 484 GNSS continuously operating reference stations with 21 concrete pillar stations that serve as the backbone for SWEREF 99 (the national reference frame). To keep the time series of these 21 fundamental stations consistent, the antennas of these pillar stations will not be changed as long as they work properly. To be able to track all new signals, such as Galileo and GPS L5 properly, starting from 2012, a second monument, a steel grid mast, was installed close to each pillar station with a newer antenna (LEIAR25.R3) and a radome (LEIT). In addition, there are other types of antennas, radomes, and monuments used in the SWEPOS network. It is therefore of interest to investigate the performance of different station designs.

We used eight GNSS stations of different designs, co-located at the Onsala Space Observatory, to estimate linear horizontal gradients. The specifications of the GNSS stations and their data processing are described in Section 2.1 and the WVR data acquisition and processing are presented in Section 2.2. In Section 3.1 we first present the agreement of the horizontal gradients between the eight GNSS stations. Thereafter, in Section 3.2 the GNSS gradients are compared to the WVR gradients. This included an assessment of the impact of the retrieval accuracy of the gradients estimated from the WVR data due to the dependence on the estimated liquid water content (LWC) from the WVR. In Section 3.3 we study the impact of using a weaker constraint for the random walk process of the GNSS gradient time series. In order to reduce the random noise we averaged GNSS gradients obtained from the different stations and compared with the WVR gradients. This is similar to what was done by Wang et al. (2024) where they averaged observations in order to reduce the noise from inexpensive co-located GNSS receivers. Discussions and conclusions are given in Sections 4 and 5, respectively.

**Table 1.** The specifications of the GNSS stations.

| Station | Antenna | Radome | Receiver | Pillar | Height [m][a] | Distance [m][b] |
|---------|---------|--------|----------|--------|------------|--------------|
| ONSA | AOADM/M_B | OSOD | Sept Polarx5tr | 1 m concrete | 10.2 | 11 |
| ONS1 | LEIAR25.R3 | LEIT | Trimble alloy | 3 m steel | 8.1 | 54 |
| OTT1 | LEIAR20 | LEIM | Trimble NetR9 | 3 m steel | 9.3 | 404 |
| OTT2 | LEIAR20 | LEIM | Trimble NetR9 | 3 m steel | 10.5 | 441 |
| OTT3 | TPSCR.G5 | TPSH | Trimble NetR9 | 3 m steel | 8.8 | 315 |
| OTT4 | LEIAR20 | LEIM | Trimble NetR9 | 3 m steel | 10.9 | 438 |
| OTT5 | TPSCR.G5 | TPSH | Trimble NetR9 | None | 12.4 | 420 |
| OTT6 | LEIAR20 | LEIM | Trimble NetR9 | 2 m steel | 11.0 | 474 |

[a] The heights are referenced to the geoid.

[b] The distance from the location of the WVR.

## 2 Datasets

### 2.1 GNSS

The eight GNSS stations and their locations are shown in Figures 1 and 2. Table 1 summarizes the specifications of the stations. ONSA is one of the 21 SWEPOS fundamental stations, but with a 1 m concrete pillar, instead of 3 m, and a microwave absorbing plate below the antenna. It is also a station in the IGS and EUREF networks. ONS1 is one of the 21 secondary stations which are installed close to each pillar station and has a 3 m steel mast. An additional 6 GNSS stations have been deployed close to the Onsala twin telescopes (OTT), used for geodetic very long baseline interferometry (VLBI), OTT1, OTT2, OTT3, OTT4, and OTT6 have an installation similar to ONS1. The design of the OTT5 station is made significantly different from the others. The antenna is mounted directly above the bedrock with a plate of microwave absorbing material below. All eight GNSS stations have hemispheric radomes. Many studies have shown that a hemispheric radome design is preferred in order to minimize errors in the estimated zenith wet delay (ZWD) and coordinates, see e.g., Emardson et al. (2000) and Ning et al. (2011). All six OTT stations underwent a receiver change in February 2022 to enable tracking of all Beidou satellites. To ensure consistent data and satellite geometry across all eight stations, we decided to include data acquired from March 2022 to December 2023.

The GNSS data processing was carried out using GipsyX v.2.0 (Bertiger et al., 2020) with the precise point positioning (PPP) strategy (Zumberge et al., 1997). The processing uses ionospheric free linear combinations formed by acquired GNSS phase-delay observations, with a sampling rate of 30 s. The output included station coordinates, clock biases, and atmospheric parameters, i.e., ZTD and linear horizontal gradients. The final multi-GNSS orbit and clock products were provided by the center for orbit determination in Europe (CODE) (Prange et al., 2020). We used an elevation cutoff angle of $10°$ to follow

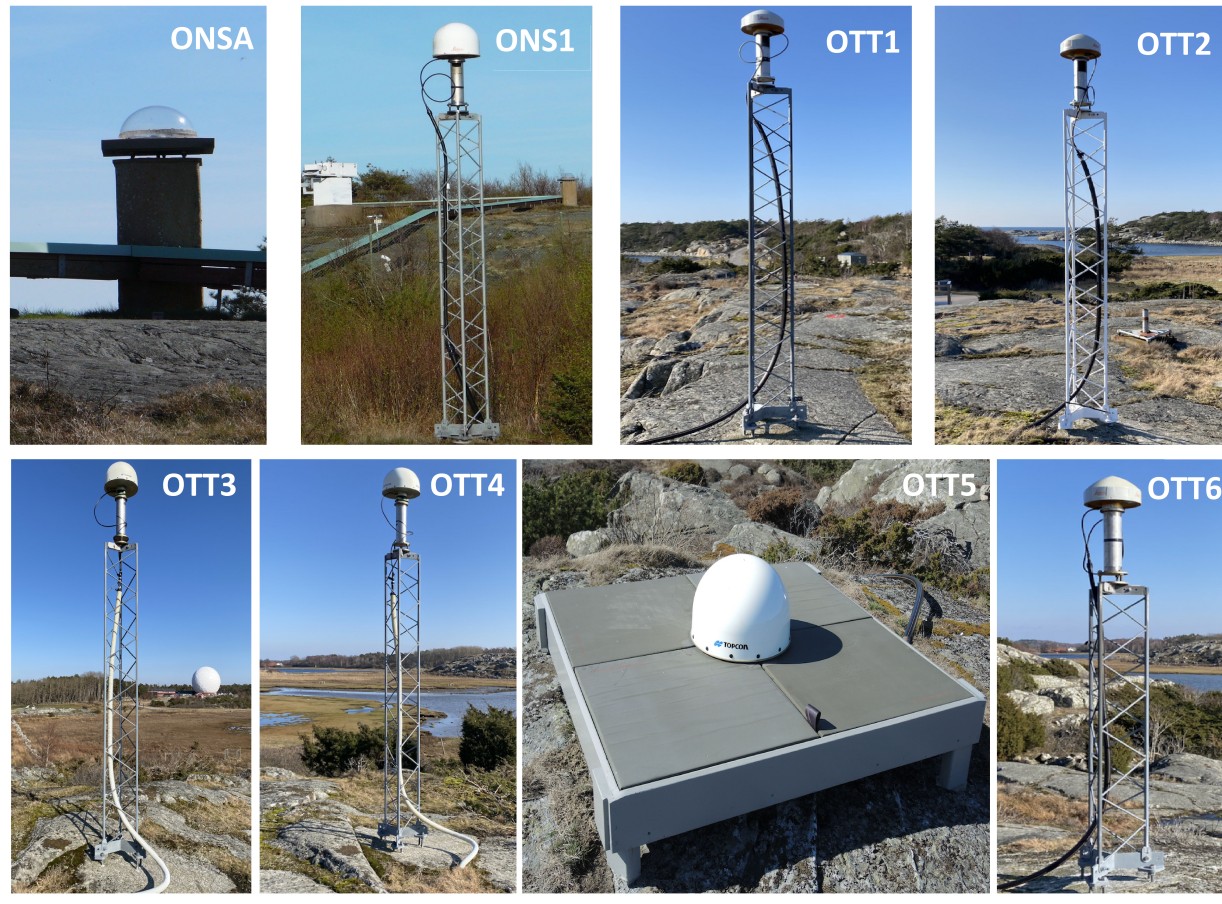

**Figure 1.** The eight GNSS stations (from left to right) are ONSA, ONS1, OTT1 to OTT6.

the finding from Ning and Elgered (2021) where the best agreement between the GNSS gradients and WVR gradients was obtained by the GNSS solution using an elevation cutoff angle of $10°$. Furthermore, the absolute calibration of the phase centre variations (PCVs) for all antennas (from the file igs14_2136.atx) was implemented (Schmid et al., 2007). The slant delays were mapped to the zenith using the vienna mapping function 1 (VMF1) (Boehm et al., 2006). The ZTD and the horizontal delay gradients were estimated every 5 min using a random walk model with constraints for the standard deviations (SD) of $10\,\mathrm{mm}\sqrt{\mathrm{h}}^{-1}$ and $0.3\,\mathrm{mm}\sqrt{\mathrm{h}}^{-1}$, respectively. We did not apply any elevation-dependent weighting to the GNSS observations based on the conclusion given by Elgered et al. (2019) where they found that the GNSS solution without weighting gives a better agreement with the WVR gradients compared to the solution with elevation-dependent weighting. The constraint value used for the ZTD was given by Jarlemark et al. (1998) where they found a temporal variability in the wet delay, derived from 71 days of WVR measurements, varying in the interval $3$–$22\,\mathrm{mm}\sqrt{\mathrm{h}}^{-1}$ at the Onsala site. In GipsyX, the default SD for gradients is $0.3\,\mathrm{mm}\sqrt{\mathrm{h}}^{-1}$. In order to assess the impact on the gradient estimation, we have also applied a weaker constraint of $2.0\,\mathrm{mm}\sqrt{\mathrm{h}}^{-1}$.

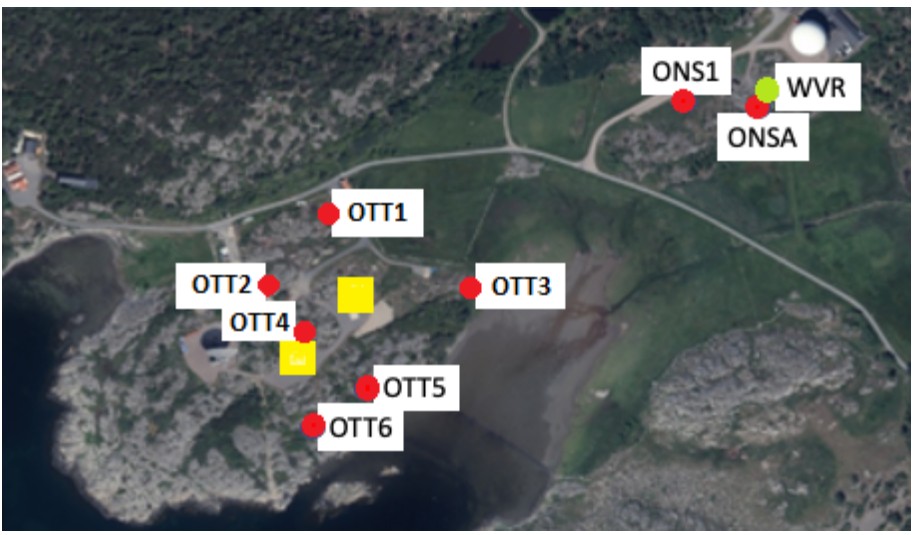

**Figure 2.** The locations of the eight GNSS stations (red circles), the WVR (green circle), and the twin telescopes used for geodetic VLBI (yellow squares). Distances to the WVR are given in Table 1.

We used the model presented by Bar-Sever et al. (1998) for the gradient estimation:

$$S(\epsilon, \phi) = m(\epsilon)\left[Z + cot(\epsilon)(G_n \, cos(\phi) + G_e \, sin(\phi))\right] \tag{1}$$

where $S(\epsilon, \phi)$ is a slant delay for a certain elevation and azimuth angle; $Z$ and $m(\epsilon)$ are the zenith delay and the elevation mapping function; $G_n$ and $G_e$ are the north and east horizontal gradient, respectively. Given that the WVR only provides the wet gradient while the GNSS gradient is the sum of the wet and the hydrostatic components, we need to subtract the hydrostatic component from the GNSS estimates to be able to compare with the WVR gradients. The horizontal hydrostatic gradients used were calculated by the VMF Data Server (2024) which is based on the ERA5 numerical weather model. Since hydrostatic gradients are mainly caused by horizontal gradients in the pressure, they have much less short-term variability (Elgered et al., 2019). During the study period the hydrostatic east gradient ranged from $-1.1$ to $0.7$ mm, with a mean of $-0.1$ mm and an SD of $0.3$ mm. The hydrostatic north gradient ranged from $-1.2$ to $0.5$ mm, with a mean of $-0.3$ mm and an SD of $0.3$ mm.

### 2.2 Water vapour radiometer

The WVR was designed in order to provide independent estimates of the wet propagation delays for space geodetic applications. It is fully steerable and measures the emission from the sky in two channels centered at 20.65 GHz and 31.63 GHz. These observations are used to derive the equivalent ZWD and the liquid water content (LWC). More detailed specifications are given by Elgered et al. (2019).

The WVR has a maximum distance of 474 m to the GNSS stations (see Table 1). The height difference to ONSA is less than 0.5 m. Because the maximum height difference between the WVR and all the other GNSS stations is 2.2 m, no attempt was made to make any model based correction as a function of the antenna height of the GNSS stations.

The WVR observations on the sky are illustrated in Fig. 3. Two different schemes were used. From 2022 the observations were scheduled in a 2 min long cycle with the ambition to sample the whole atmosphere at elevation angles from 30°. For this work, however, the highest temporal resolution is 5 min. Furthermore, from 22 August 2023 the observational cycle was changed to a much more dense sampling of the sky during a cycle of 5 min. A disadvantage of this WVR, when it is used to estimate horizontal gradients, is that the azimuth interval was limited to 0°–180°, meaning that all observations to the west

were acquired for elevation angles > 90°. This limitation was used because of earlier operational problems with tension in cables. Another disadvantage is that the two channels have independent antenna pointing and therefore pointing errors could be different. This makes the estimated gradients in the east direction extra sensitive to pointing errors in the elevation angle. We will come back to this when the results are discussed,

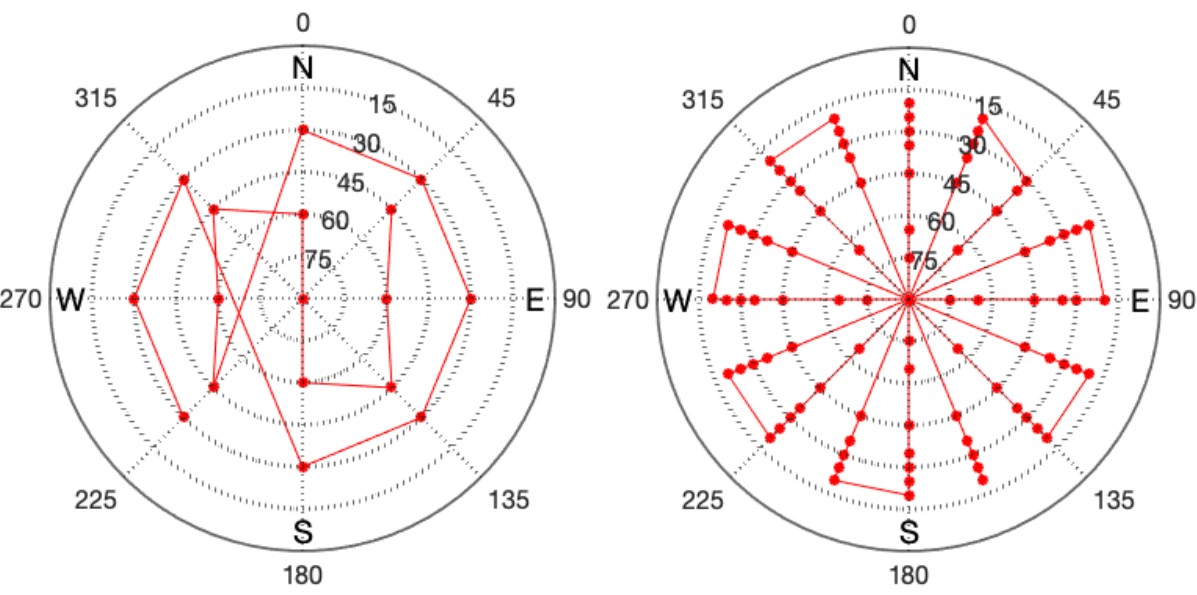

**Figure 3.** From 1 March 2022 to 22 August 2023 the WVR made 17 observations in a 2 min cycle (left). Thereafter, a 5 min cycle was used with 90 observations (right), and from 15 November an additional 7 observations were added, to tune the cycle to 5 min, within a couple of seconds. The zenith point is only measured once in each cycle. Due to blockage of a radio telescope there was no observation in the direction azimuth = 45° and elevation = 20°.

  A four-parameter model was used to estimate a mean ZWD, a linear trend in the ZWD, and east and north linear horizontal

gradients over a 5 min period (Davis et al., 1993). No constraint between estimates in adjacent periods was applied. Before the final processing of the WVR raw data, periods of rain, indicated by a rain sensor at the site, were identified. Data were removed from 5 min before the rain started until at least 15 min after it stopped. The amount of data removed after each rain event was based on a subjective inspection of the ZWD time series. Depending on the weather conditions, the time needed for water drops to disappear from the WVR feeds varies.

In order to apply the four-parameter model, we required that each observation used had an estimated equivalent zenith LWC $< 0.7$ mm. We also required that at least 30 observations were available in each 5 min interval during the period from March 2022 to August 2023. From September 2023, when the sample rate had been increased, this requirement was increased to at least 75 observations.

A second WVR dataset was also produced where all individual WVR observations resulting in a LWC $> 0.05$ mm were ignored. The reason for applying this strict editing is to have as accurate estimates as possible in the comparisons to the gradient results from the eight GNSS stations. Since the WVR estimates are fully independent of the corresponding estimates from the GNSS data they form a suitable reference dataset.

## 2.3 The different datasets used

Horizontal gradients can change rapidly and there are data gaps. Therefore, it is important to synchronize the datasets from the stations being compared to ensure that the results obtained only use data points when observations are available from all stations. In addition to the GNSS only datasets, we use four different datasets when synchronized with the WVR. Figure 4 shows the number of available data points from GNSS and WVR when synchronized.

Table 2 summarizes the statistics of the formal error for the estimated gradients. As expected, applying a weak constraint in the GNSS data processing increases the formal error of the gradients. ONSA has the smallest formal errors. It is equipped with microwave-absorbing material, below the antenna above the metal plate used for the antenna mounting, which likely reduces unwanted multipath effects. It is also interesting to note that OTT5 has comparable formal errors to the other stations, although the antenna is mounted directly above the bedrock. The WVR data exhibit slightly larger formal errors for both the east and the north gradients when the larger threshold (0.7 mm) for the LWC is used. The formal errors of the WVR gradients are slightly reduced when the amplitude of the gradient is less than 0.5 mm but this is not the case for the relative formal errors.

To provide some details about the estimated gradients, Figure 5 shows the time series of the gradients from 00 UT on 1 August to 24 UT on 4 August, 2022. The GNSS gradients are shown for the two different constraints used. When a weak constraint is applied, there is an improvement in the tracking of rapid variations in the gradients. However, the overall gradients given by a weak constraint also show greater scatter compared to those given by a strong constraint. This issue is discussed in detail in Section 3.3.

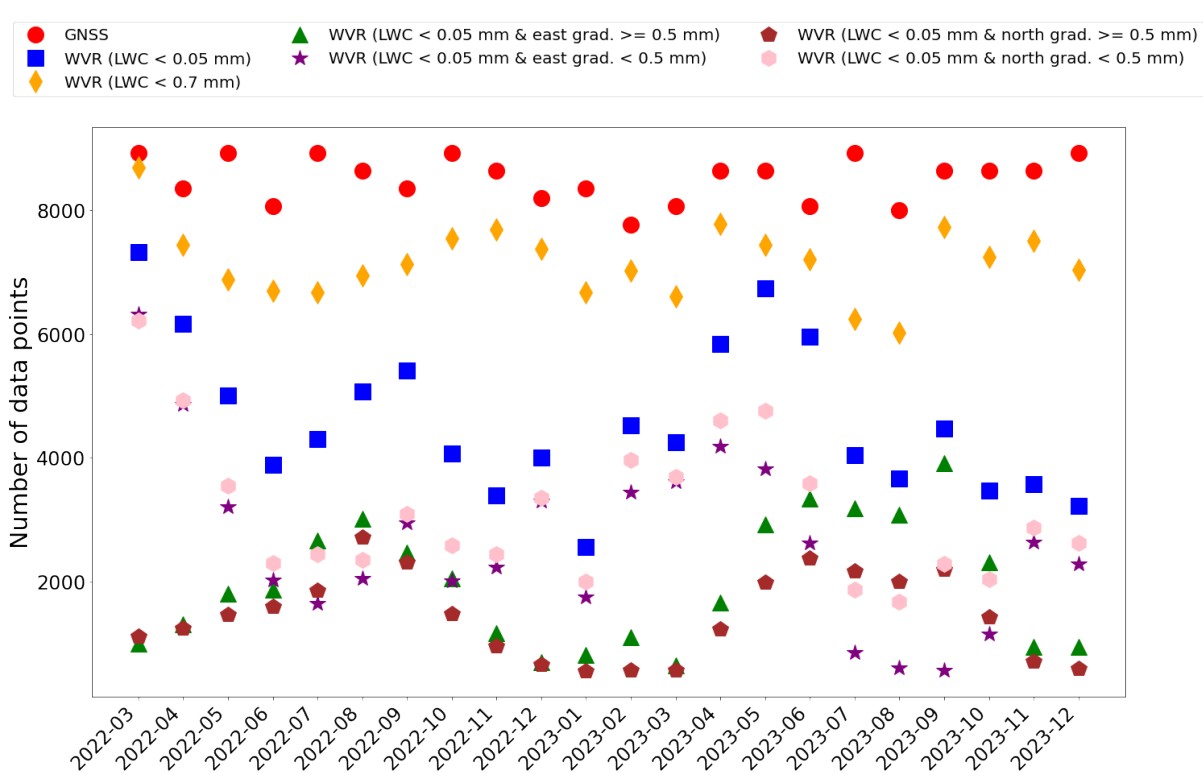

**Figure 4.** Number of simultaneous data points for all 8 GNSS stations is 187,269; available data points while LWC < 0.05 mm or < 0.7 mm is 100,938 and 157,670, respectively. The smaller dataset, when LWC < 0.05 mm, is divided into four datasets. When the east gradient amplitude $\geq$ 0.5 mm or < 0.5 mm it results in 42,830 and 58,108 data points, respectively. The corresponding numbers of data points when the north gradient amplitude $\leq$ 0.5 mm or < 0.5 mm are 31,707 and 69,231, respectively.

**Table 2.** The formal error, mean and SD (in parentheses), for GNSS and WVR gradients.

| Station | Constraint of 0.3 mm$\sqrt{h}^{-1}$ | | Constraint of 2.0 mm$\sqrt{h}^{-1}$ | |
|---|---|---|---|---|
| | EG | NG | EG | NG |
| | [mm] | [mm] | [mm] | [mm] |
| ONSA | 0.14 (0.03) | 0.15 (0.03) | 0.30 (0.08) | 0.32 (0.08) |
| ONS1 | 0.16 (0.03) | 0.18 (0.03) | 0.33 (0.12) | 0.35 (0.13) |
| OTT1 | 0.16 (0.03) | 0.18 (0.03) | 0.36 (0.12) | 0.39 (0.12) |
| OTT2 | 0.17 (0.03) | 0.17 (0.03) | 0.36 (0.09) | 0.38 (0.10) |
| OTT3 | 0.16 (0.03) | 0.16 (0.03) | 0.35 (0.13) | 0.36 (0.13) |
| OTT4 | 0.17 (0.03) | 0.19 (0.03) | 0.37 (0.10) | 0.41 (0.12) |
| OTT5 | 0.16 (0.03) | 0.17 (0.03) | 0.37 (0.13) | 0.38 (0.14) |
| OTT6 | 0.18 (0.03) | 0.19 (0.03) | 0.40 (0.13) | 0.43 (0.13) |
| *Each WVR gradient is independent on adjacent values in the estimation process* | | | | |
| WVR, LWC $< 0.7$ mm | | | 0.10 (0.12) | 0.09 (0.11) |
| WVR, LWC $< 0.05$ mm | | | 0.07 (0.04) | 0.06 (0.04) |
| WVR, LWC $< 0.05$ mm + east grad. ampl.$\geq 0.5$ mm | | | 0.08 (0.05) | 0.07 (0.04) |
| WVR, LWC $< 0.05$ mm + north grad. ampl.$\geq 0.5$ mm | | | 0.08 (0.05) | 0.08 (0.04) |
| WVR, LWC $< 0.05$ mm + east grad. ampl.$< 0.5$ mm | | | 0.06 (0.03) | 0.06 (0.03) |
| WVR, LWC $< 0.05$ mm + north grad. ampl.$< 0.5$ mm | | | 0.06 (0.03) | 0.06 (0.03) |

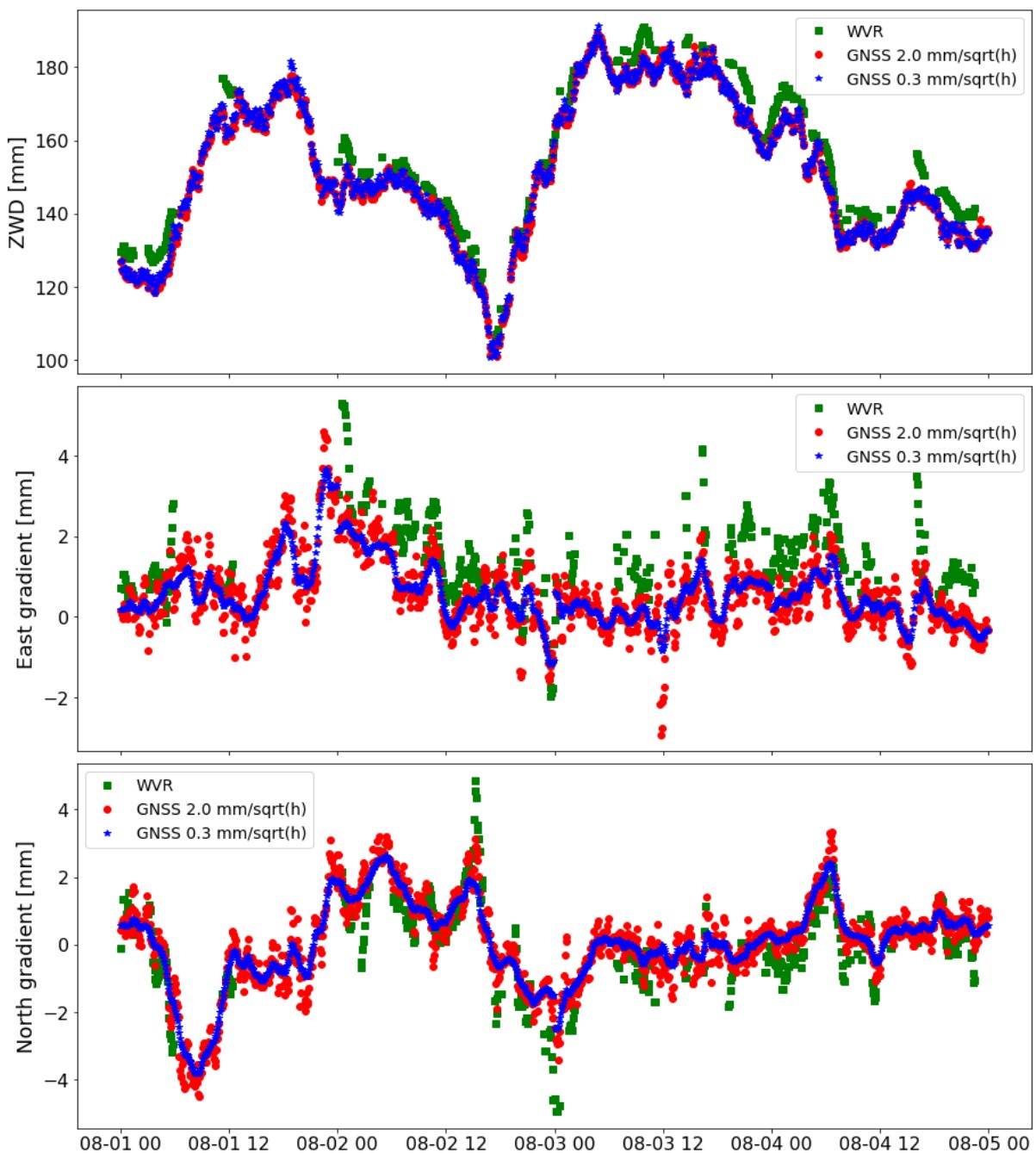

**Figure 5.** An example of gradient time series from the WVR and from two GNSS solutions for the ONSA station, using different constraints, 0.3 and 2.0 mm$\sqrt{h}^{-1}$.

## 3 Comparison results

### 3.1 GNSS gradients vs GNSS gradients

Initially, we assess the gradients derived from the eight GNSS stations in comparison with one another. We expect a strong agreement, given the similarity in the atmospheric sampling across the all stations and the shared presence of various error sources among them. On the other hand, such pairwise comparison from several co-located stations can also reveal errors from a specific station. The results in terms of root mean square (RMS) differences and correlations are shown in Table 3. The RMS differences are dominated by the standard deviations meaning that the biases are small, of the order of 0.1 mm or less, which indicates that there is no permanent severe multipath effects in a specific direction at any of the eight stations. During the study period, the mean value of the east gradients, for the ONSA station, is $-0.01$ mm with an SD of 0.54 mm, while the north gradients have a mean of $-0.06$ mm with an SD of 0.50 mm. This illustrates that the bias is negligible compared to the variability of the gradients. Furthermore, We observe rather consistent agreements across all stations. The OTT4 station does, however, show slightly larger RMS differences and lower correlations in the north gradient as a consequence from less observations in the south-south-west direction caused by the blockage of the VLBI telescope (see Fig. 2).

The OTT5 antenna, mounted directly above the bedrock, gives consistent agreement with other stations showing no clear impact from the environment, such as multipath. This is probably due to the plate of microwave absorbing material placed right below the antenna.

We note that the agreements between ONSA/ONS1 and between the six OTT stations are not significantly better indicating that the longer distances of a few hundred metres are not important when observations are averaged over 5 min intervals.

### 3.2 GNSS gradients vs WVR gradients

In order to investigate the influence of the amount of liquid water in the atmosphere on the retrieval accuracy of the gradients using the WVR, we have applied two LWC threshold values ($< 0.05$ and $< 0.7$ mm) for which individual WVR observations to include in each 5 min estimate of the four model parameters. Table 4 shows the results from the comparison between the GNSS and the WVR gradients. We chose to present biases, standard deviations, and RMS differences. In this case the biases cannot be ignored, especially in the east direction. The biases between the WVR and GNSS are approximately equal to the mean values of the WVR gradients. We expect the true values of the mean gradients to be close to zero over a time period of almost two years. This is also what we see from the GNSS and ERA5 results. We identify two likely sources for the WVR bias. One is a misalignment in the azimuth bearing and the other is errors in the elevation angle. It is discussed further in the next section.

Although the differences are small, we see that the ONSA station has the best, while the OTT4 station exhibits the worst agreement, with the WVR estimates for both datasets and for both the east and the north gradients. In the following, all comparisons are made using WVR gradients obtained from observations with an LWC $< 0.05$ mm in order to have as high a quality as possible.

**Table 3.** Correlation coefficients (upper right triangle) and RMS differences in mm (lower left triangle) for horizontal gradients over the whole time period. The best and the worst agreements are highlighted by bold and italic font, respectively.

| Station | ONSA | ONS1 | OTT1 | OTT2 | OTT3 | OTT4 | OTT5 | OTT6 |
|---|---|---|---|---|---|---|---|---|
| | | | | *East gradient* | | | | |
| ONSA | – | 0.92 | 0.93 | 0.92 | **0.94** | 0.92 | 0.93 | 0.91 |
| ONS1 | 0.24 | – | 0.91 | *0.90* | 0.92 | 0.91 | 0.93 | 0.91 |
| OTT1 | 0.23 | 0.25 | – | 0.93 | 0.93 | 0.93 | 0.92 | 0.91 |
| OTT2 | 0.25 | *0.27* | 0.23 | – | 0.93 | 0.93 | 0.92 | *0.90* |
| OTT3 | 0.23 | *0.27* | 0.23 | 0.26 | – | 0.92 | **0.94** | 0.91 |
| OTT4 | 0.25 | *0.27* | 0.23 | 0.24 | 0.25 | – | 0.92 | 0.91 |
| OTT5 | 0.23 | 0.24 | 0.23 | 0.26 | 0.22 | 0.25 | – | **0.94** |
| OTT6 | 0.25 | 0.25 | 0.25 | *0.27* | 0.26 | 0.26 | **0.21** | – |
| | | | | *North gradient* | | | | |
| ONSA | – | 0.89 | 0.89 | 0.90 | **0.92** | 0.86 | 0.91 | 0.89 |
| ONS1 | 0.26 | – | 0.87 | 0.88 | 0.90 | *0.85* | 0.90 | 0.88 |
| OTT1 | 0.26 | 0.29 | – | 0.91 | 0.91 | 0.88 | 0.89 | 0.88 |
| OTT2 | 0.25 | 0.28 | 0.25 | – | **0.92** | 0.89 | 0.90 | 0.88 |
| OTT3 | 0.22 | 0.25 | 0.23 | 0.23 | – | 0.89 | **0.92** | 0.90 |
| OTT4 | 0.29 | *0.31* | 0.28 | 0.26 | 0.27 | – | 0.86 | *0.85* |
| OTT5 | 0.23 | 0.24 | 0.26 | 0.25 | **0.21** | 0.29 | – | 0.91 |
| OTT6 | 0.27 | 0.27 | 0.29 | 0.27 | 0.26 | *0.31* | 0.24 | – |

### 3.3 GNSS gradients estimated with different constraints

We have applied two different constraints, 0.3 or 2.0 mm$\sqrt{h}^{-1}$, for the SD in the random walk model for the GNSS gradients. The comparison results to the WVR are presented in Table 5. As already noted in Section 2.3, using a weak constraint enhances the GNSS data tracking large gradients at the cost of larger formal errors. This also affects the agreement between GNSS and WVR gradients, which is confirmed in the table.

Table 5 also presents the agreement results for the two datasets, categorized by the absolute amplitude of gradients derived from the WVR: $< 0.5$ mm or $> 0.5$ mm for the east and north gradients, respectively. As expected, these results show that both the correlation and the SD increase when gradients are large. When gradients are very small, there is almost no correlation to be found and the SD shall be just the random errors. When gradients are larger there is also a contribution to the differences from the different sampling of the sky.

To compare the two solutions using different constraints we carried out averaging of the gradients from the individual GNSS stations. As pointed out by Ning et al. (2016) errors in GNSS measurements are random or systematic and random errors are uncorrelated across different stations, and can be reduced through averaging. The procedure starts with comparison between

**Table 4.** Agreement between the horizontal gradients from mutli-GNSS and the WVR, estimated using two different maximum values for the LWC from the WVR. The best and the worst agreements in each dataset are highlighted by bold and italic font, respectively.

| Station | Bias[a] | | SD | | RMS | | Correlation | |
|---|---|---|---|---|---|---|---|---|
| | EG | NG | EG | NG | EG | NG | EG | NG |
| | [mm] | [mm] | [mm] | [mm] | [mm] | [mm] | | |
| *LWC < 0.05 mm* | | | | | | | | |
| ONSA | −0.41 | 0.14 | **0.55** | **0.48** | 0.69 | **0.50** | **0.71** | **0.69** |
| ONS1 | −0.44 | 0.15 | 0.57 | 0.51 | 0.72 | 0.53 | 0.67 | 0.64 |
| OTT1 | −0.41 | 0.09 | 0.57 | 0.52 | 0.70 | 0.53 | 0.69 | 0.63 |
| OTT2 | −0.46 | 0.15 | *0.58* | 0.50 | *0.74* | 0.52 | 0.67 | 0.65 |
| OTT3 | −0.33 | 0.12 | 0.57 | 0.49 | **0.66** | **0.50** | 0.69 | 0.67 |
| OTT4 | −0.40 | 0.12 | *0.58* | *0.53* | 0.70 | *0.54* | *0.66* | *0.62* |
| OTT5 | −0.36 | 0.12 | 0.57 | 0.49 | 0.67 | **0.50** | 0.68 | 0.67 |
| OTT6 | −0.42 | 0.19 | *0.58* | 0.51 | 0.72 | *0.54* | *0.66* | 0.65 |
| *LWC < 0.7 mm* | | | | | | | | |
| ONSA | −0.50 | 0.15 | **0.70** | **0.61** | 0.86 | **0.63** | **0.67** | **0.65** |
| ONS1 | −0.53 | 0.17 | 0.72 | 0.64 | 0.89 | *0.66* | 0.64 | 0.61 |
| OTT1 | −0.49 | 0.12 | 0.71 | 0.64 | 0.86 | 0.65 | 0.64 | 0.60 |
| OTT2 | −0.54 | 0.17 | 0.72 | 0.63 | *0.90* | 0.65 | *0.63* | 0.62 |
| OTT3 | −0.42 | 0.14 | 0.71 | 0.62 | **0.82** | 0.64 | 0.65 | 0.64 |
| OTT4 | −0.49 | 0.14 | *0.73* | *0.65* | 0.88 | *0.66* | *0.63* | *0.58* |
| OTT5 | −0.46 | 0.14 | 0.72 | 0.62 | 0.85 | 0.64 | 0.64 | 0.64 |
| OTT6 | −0.50 | 0.21 | *0.73* | 0.63 | 0.88 | *0.66* | *0.63* | 0.61 |

[a] The bias is defined as GNSS−WVR.

WVR gradients and gradients from a single station (ONSA). Thereafter, we took the gradients from the ONSA station and calculated an average with the ones from the ONS1 station. This new set of average gradients was then compared to the WVR gradients to assess their agreement. Then we incorporated the gradients from the OTT1 station into the previously averaged data from ONSA and ONS1, creating a new combined average and compared to WVR gradients again. We continued this iterative process by gradually adding the gradients from additional stations, one by one, each time recalculating the average and comparing it with the WVR gradients. This process was repeated until we had included the gradients from all eight GNSS stations. The result obtained when averaging all eight stations is the entry called "fusion" in Table 5. Figure 6 illustrates the agreement obtained when another GNSS station is added to the calculation of the mean value. We notice that while the averaging improve the agreement with the WVR gradients when applying the weak constraint, the effect for the strong constraint

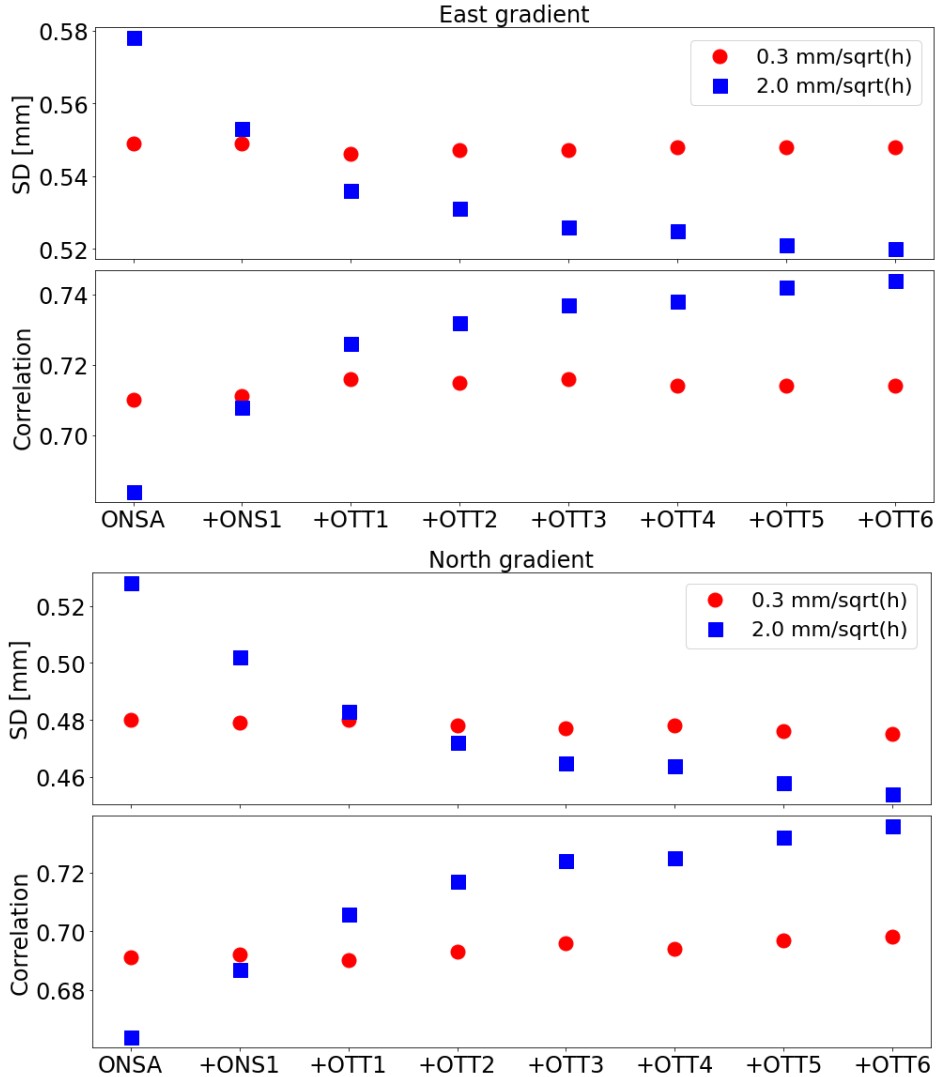

**Figure 6.** Averaged GNSS gradients vs WVR gradients.

is hardly significant. Our interpretation is that the use of the week constraint results in more random errors whereas the errors present in the solution using a strong constraint is dominated by systematic effects.

Finally, since the presence of gradients varies a lot with the weather, and therefore also the seasons, the result for each month is of interest. These correlations and the standard deviations are presented in Figure 7. The monthly results are consistent with the overall result for the whole dataset, although the level of agreement varies with the seasons. The results indicate that for months with large gradients, such as July and August 2022 and September 2023, the weak constraints yield a better correlation with the WVR data. However, for months with small gradients, worse correlations are seen due to the dominance of the noise introduced by weak constraints and the smaller dynamic range of the gradients. This pattern is also observed for

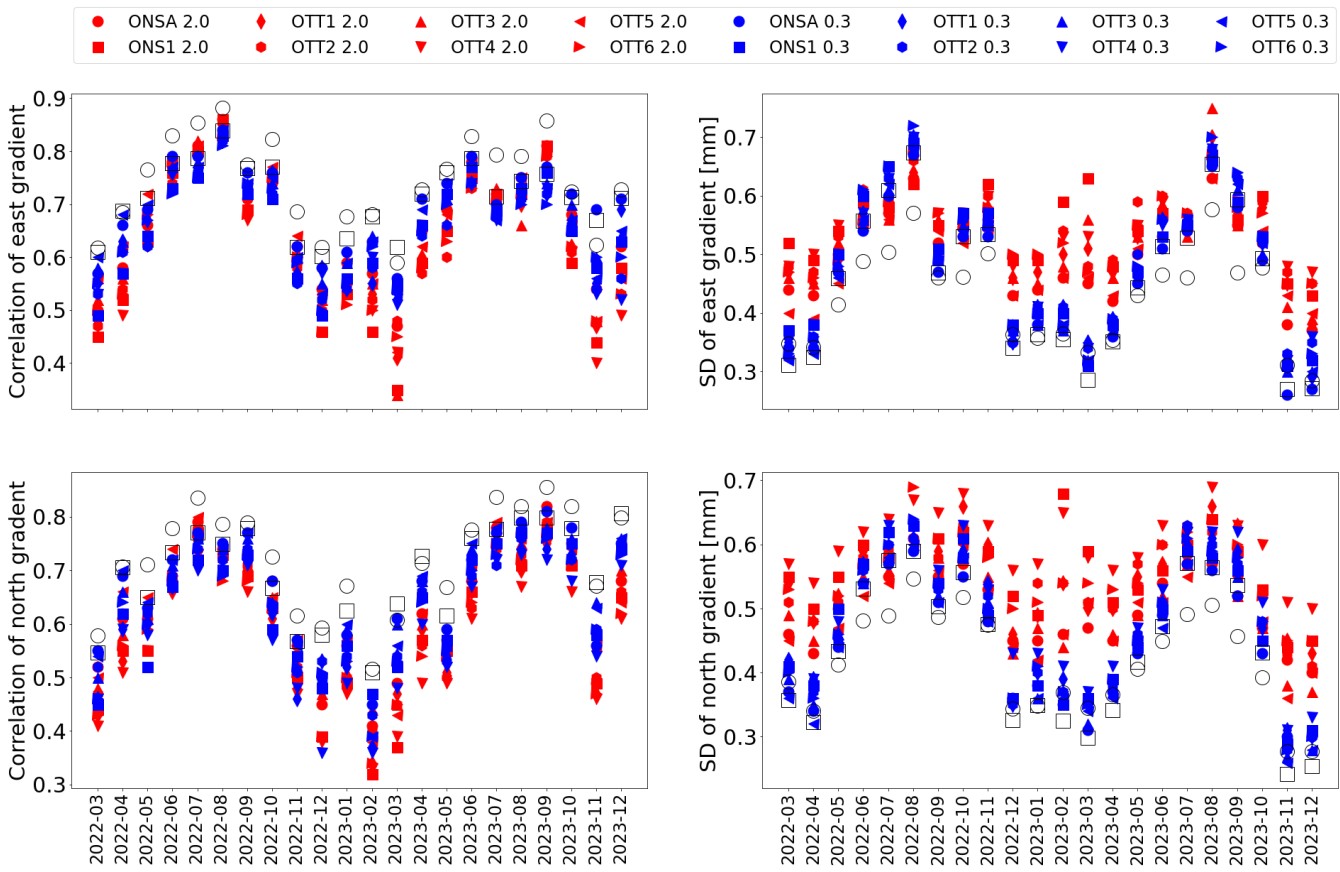

**Figure 7.** The correlation (left) and standard deviation (right) between the GNSS and the WVR gradients for each month when two different constraints are applied in the GNSS data processing. The values from the averaged GNSS gradients are shown by black circles for the constraint of 2.0 mm$\sqrt{\mathrm{h}}^{-1}$ and by black squares for the constraint of 0.3 mm$\sqrt{\mathrm{h}}^{-1}$.

the SD results, where weak constraints produce smaller or similar values compared to strong constraints in months with large gradients. Conversely, in months with small gradients, weak constraints result in larger SD as they introduced more noise in the GNSS gradients.

**Table 5.** Agreement between the horizontal gradients from mutli-GNSS and WVR. The best and the worst agreements are highlighted by bold and italic font, respectively.

| Station | Constraint of 0.3 mm$\sqrt{h}^{-1}$ | | | | Constraint of 2.0 mm$\sqrt{h}^{-1}$ | | | |
|---|---|---|---|---|---|---|---|---|
| | Correlation | | RMS | | Correlation | | RMS | |
| | EG | NG | EG | NG | EG | NG | EG | NG |
| | | | [mm] | [mm] | | | [mm] | [mm] |
| *All gradients* | | | | | | | | |
| ONSA | 0.71 | 0.69 | 0.69 | 0.50 | 0.68 | 0.66 | 0.71 | 0.55 |
| ONS1 | 0.67 | 0.64 | 0.72 | 0.53 | *0.64* | 0.61 | *0.77* | 0.61 |
| OTT1 | 0.69 | 0.63 | 0.70 | 0.53 | 0.66 | 0.62 | 0.73 | 0.59 |
| OTT2 | 0.67 | 0.65 | 0.74 | 0.52 | 0.65 | 0.63 | *0.77* | 0.59 |
| OTT3 | 0.69 | 0.67 | **0.66** | 0.50 | 0.66 | 0.65 | 0.68 | 0.55 |
| OTT4 | 0.66 | 0.62 | 0.70 | 0.54 | *0.64* | *0.59* | 0.76 | *0.63* |
| OTT5 | 0.68 | 0.67 | 0.67 | 0.50 | 0.67 | 0.66 | 0.69 | 0.54 |
| OTT6 | 0.66 | 0.65 | 0.72 | 0.54 | *0.64* | 0.62 | 0.75 | 0.61 |
| Fusion | 0.71 | 0.70 | 0.68 | 0.50 | **0.74** | **0.74** | **0.66** | **0.47** |
| *WVR gradients ≥ 0.5 mm* | | | | | | | | |
| ONSA | 0.78 | 0.77 | 0.94 | 0.85 | 0.78 | 0.77 | 0.92 | 0.85 |
| ONS1 | 0.75 | 0.74 | *1.00* | 0.90 | 0.75 | 0.75 | 0.98 | 0.90 |
| OTT1 | 0.75 | 0.75 | 0.96 | 0.87 | 0.75 | 0.75 | 0.93 | 0.87 |
| OTT2 | *0.74* | 0.74 | *1.00* | *0.91* | 0.75 | 0.74 | 0.98 | *0.91* |
| OTT3 | 0.76 | 0.75 | 0.91 | 0.83 | 0.76 | 0.76 | **0.87** | **0.81** |
| OTT4 | *0.74* | 0.74 | 0.98 | 0.88 | *0.74* | 0.74 | 0.95 | 0.89 |
| OTT5 | 0.75 | 0.75 | 0.95 | 0.85 | 0.76 | 0.76 | 0.90 | 0.82 |
| OTT6 | *0.74* | *0.73* | *1.00* | 0.89 | *0.74* | 0.74 | 0.96 | 0.88 |
| Fusion | 0.77 | 0.77 | 0.95 | 0.86 | **0.82** | **0.81** | **0.87** | **0.81** |
| *WVR gradients < 0.5 mm* | | | | | | | | |
| ONSA | 0.36 | 0.31 | 0.40 | 0.34 | 0.31 | 0.28 | 0.51 | 0.47 |
| ONS1 | 0.32 | 0.25 | 0.42 | 0.37 | *0.27* | *0.23* | *0.58* | 0.54 |
| OTT1 | 0.35 | 0.24 | 0.40 | 0.38 | 0.30 | 0.24 | 0.54 | 0.52 |
| OTT2 | 0.33 | 0.27 | 0.44 | 0.36 | 0.28 | 0.25 | *0.58* | 0.52 |
| OTT3 | 0.34 | 0.28 | **0.36** | 0.35 | 0.29 | 0.27 | 0.50 | 0.46 |
| OTT4 | 0.32 | 0.25 | 0.41 | 0.39 | *0.27* | *0.23* | 0.56 | *0.56* |
| OTT5 | 0.35 | 0.28 | 0.37 | 0.35 | 0.31 | 0.26 | 0.47 | 0.46 |
| OTT6 | 0.34 | 0.27 | 0.40 | 0.37 | 0.29 | 0.25 | 0.55 | 0.53 |
| Fusion | **0.38** | 0.31 | 0.37 | **0.32** | **0.38** | **0.34** | 0.43 | 0.38 |

## 4 Discussion

We have estimated the linear horizontal gradients covering a nearly two-year period from eight co-located GNSS stations using data acquired from GPS, GLONASS, Galileo and BeiDou satellites. When the gradients obtained from the GNSS stations were compared to each other, we see consistent agreements across all stations, including OTT5, where the antenna is mounted directly above the bedrock. This is probably due to a plate of microwave absorbing material placed right below the antenna. A slightly worse agreement is observed for station OTT4 compared to the others due to the station's proximity to one of the geodetic VLBI telescopes, resulting in the loss of observations at low elevation angles in the south-south-east direction.

The better agreement for the east gradient is likely due to the geometry of the GNSS observations. At the latitudes of the stations, a substantial portion of the sky just north of the zenith remains unsampled. This finding aligns with the results presented in previous gradient studies (Elgered et al., 2019; Ning and Elgered, 2021; Elgered et al., 2023).

The GNSS-derived gradients were also compared to the ones obtained from a colocated WVR. The WVR gradients has its shortcomings. Although it is of secondary priority in this study, when the WVR offer independent gradient estimates in order to compare different GNSS solutions, it still deserves some attention.

One is the higher elevation cutoff angle compared to GNSS. According to a study by Ning and Elgered (2021), a higher elevation cutoff angle can lead to larger gradient estimates if the true variability of the wet delay in the sky includes higher-order terms in addition to a linear gradient. This means that the WVR data do not capture the variability over a larger part of the sky, resulting in a potential overestimation of the linear gradients.

Another WVR problem is errors in the elevation angle and their impact on the estimated gradients. One source to this is a misalignment of the azimuth bearing which we estimate to be at the most $0.1°$. The corresponding error in the gradient will increase proportionally with the ZWD, and it will increase with a decreasing elevation cutoff angle. Therefore, when the ZWD varies, it will appear as a systematic varying bias in the GNSS comparisons. For example, a ZWD of 100 mm, a cutoff angle of $30°$, and an angle error of $0.1°$ result in a gradient bias of the order of 0.3 mm. When the cutoff angle is decreased to $20°$ the bias is increased to 0.5 mm. For our WVR we have two additional contributions to the pointing uncertainty. One is that the observations are acquired at azimuth angles between $0°$ and $180°$ only. Observations towards the west are obtained by ordering elevation angles larger than $90°$ (see Fig. 3). The second is that the two channels have independent pointing, which will cause an extra bias in the retrieved wet delay if they are different. In summary this means that pointing errors in the elevation angles of the two channels will cause a systematic bias that will vary with the ZWD and is expected to be higher for the east gradient compared to the north gradient.

Fig. 8 illustrates the observed monthly means of the ZWD and the gradients from the GNSS fusion solution and the WVR. We see a clear correlation between the ZWD and the WVR mean gradients. We also note that the east mean gradient is increased from August 2023 when the elevation cutoff angle is lowered to $20°$. Although not perfect, this is consistent with possible pointing errors of the WVR. This means that pointing errors of the WVR are a major source of error when WVR gradients are compared to those from GNSS in an absolute sense. The error in the elevation angle of the GNSS satellites can be ignored, meaning that the monthly means of the GNSS gradients are much more accurate compared to those from the WVR.

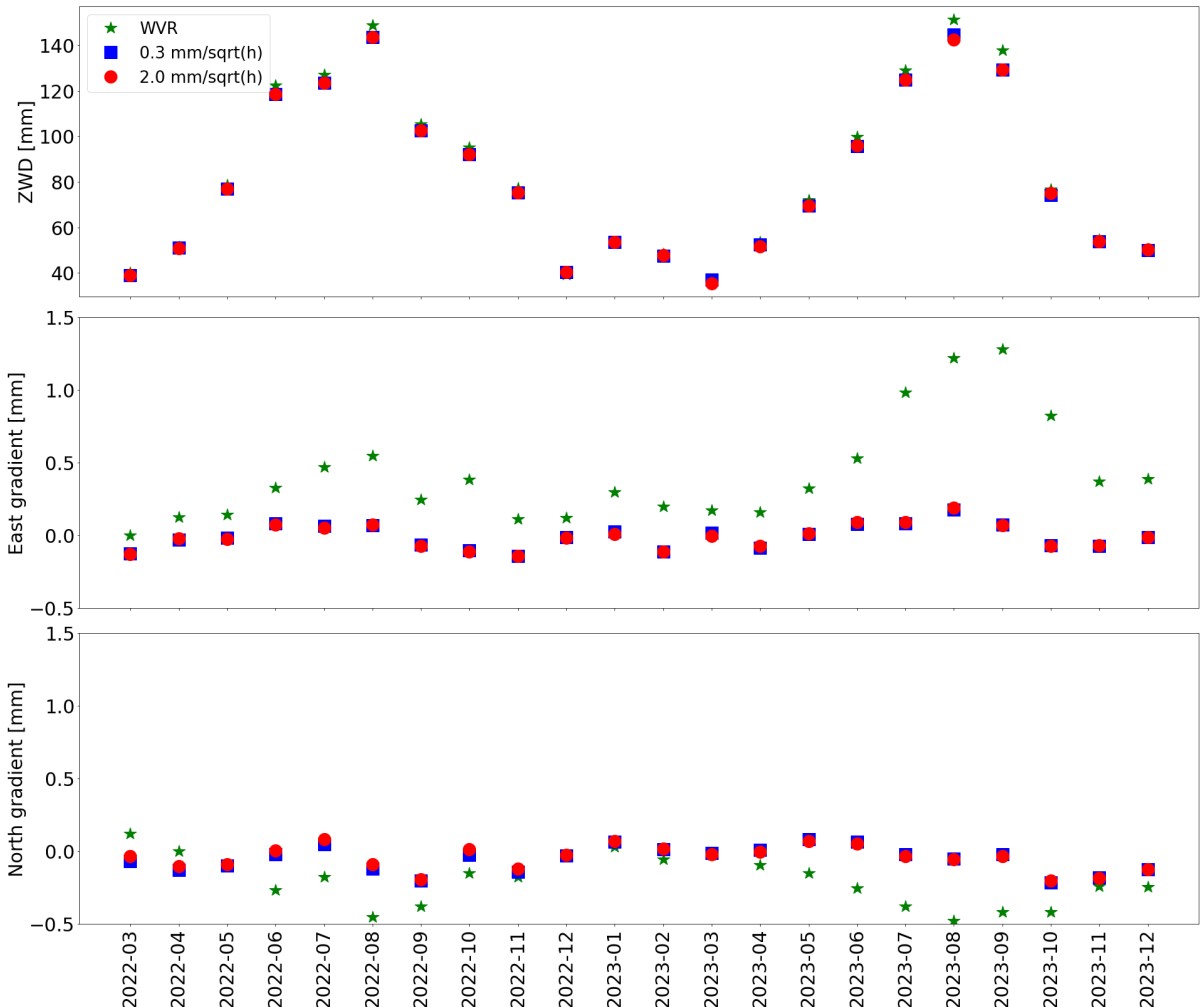

**Figure 8.** The monthly mean ZWD and gradients from the WVR and the fusion results using the eight GNSS stations.

The very small systematic variation seen in the mean GNSS east gradient may be explained by the location at the coast line. During the summer, the air over the sea is colder than over land, which could lead to more water vapor over land and a positive east gradient. During the winter, the opposite is true. The variations seen in the mean GNSS north gradient are of a similar size but are not clearly seasonal. A month of data is not sufficient to rule out that the variations seen in the monthly means are not just caused by specific weather conditions, such as passages of frontal systems.

An improved agreement was obtained when using WVR data with an LWC less than 0.05 mm compared to when LWC values of up to 0.7 mm are included. However, the lower LWC threshold results in a 35 % reduction in available WVR data points. Therefore, in some applications, and depending on the LWC characteristics at the site(s) studied, it may be important

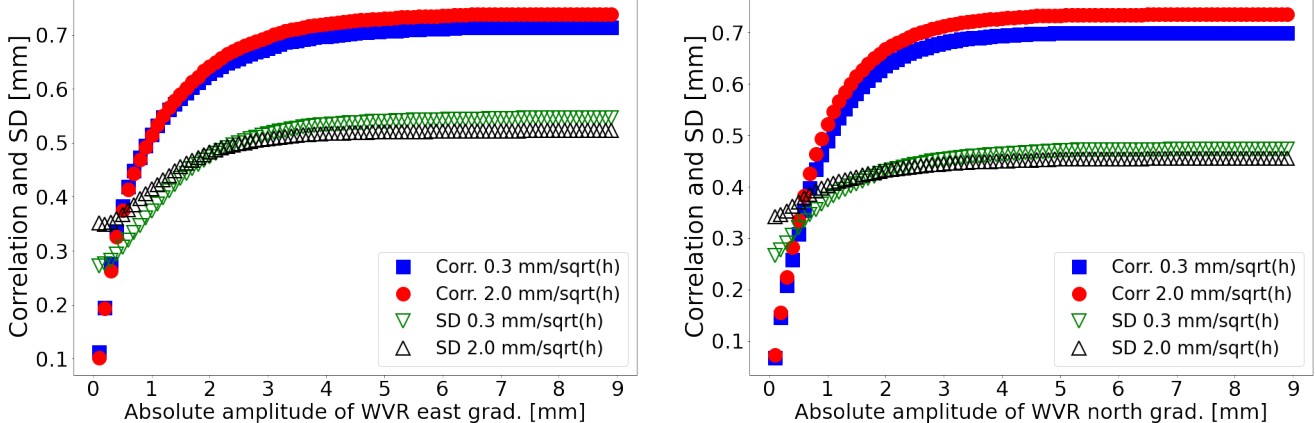

**Figure 9.** Illustration of how the correlation and the standard deviation (SD) depends on the range of observed gradients. Here correlation and SD are shown as a function of the maximum amplitude of the east (left) and north (right) gradient from the WVR.

to balance the amount of available data for tracking the variability of gradients under as many different weather conditions as possible and the accuracy of the data.

The correlation coefficients and the standard deviations (SD) between the GNSS and WVR gradients depend not only on errors of the specific techniques but also on the size and variability of the true gradients. Because different volumes of air are sampled, the estimated gradients are influenced by how well the linear model represents reality. During situations when there are no significant gradients the estimates from GNSS och WVR are their specific errors which we assume are not correlated. This is illustrated in Fig. 9. The correlation increases between the gradients from the fusion solution, using all eight stations, and the gradients from the WVR for datasets with an increasing maximum absolute value of the WVR gradients. We also see smaller SDs obtained from the GNSS solution with a strong constraint when the gradients are small. This suggests that a strong constraint is advantageous when the errors of the technique dominate the gradients.

An alternative and complementary presentation of this dependence is found in Fig. 10. Intervals (bins) are used for the values at the x axis. The amplitude intervals are defined with a bin width of $\pm 0.5$ mm. This bin width was chosen to ensure a sufficient number of data points for correlation and SD calculations at each amplitude level. For example, for the amplitude of 1 mm, the bin includes gradients with WVR amplitudes in the ranges from +0.5 to +1.5 mm and from $-1.5$ to $-0.5$ mm, with no intermediate values. Correlations and SDs are not provided for amplitudes greater than 6 mm due to the limited number of available data points. A small sample size would lead to statistically unreliable results. As shown in the figure, when only large gradients are included in the comparison, the correlation coefficient can be significantly higher and SDs also increase dramatically.

The correlations presented in this paper are similar to the findings of other studies when comparing GNSS gradients with those obtained using different techniques. For example, using one week data during the CONT14 campaign, Elgered et al. (2019) reported correlation coefficients of approximately 0.6 and 0.7 for north and east gradients between GNSS-derived

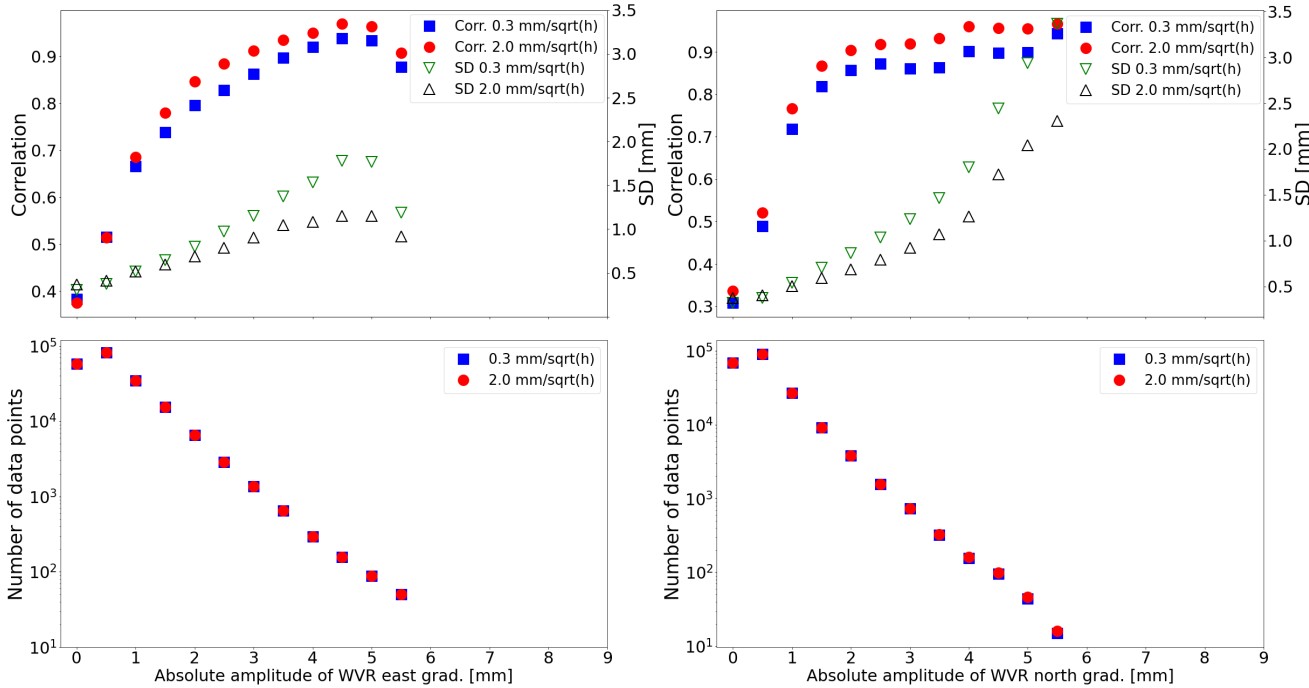

**Figure 10.** Illustration of how the correlation and SD depends on the size of the gradients observed by the WVR. Here the correlation and SD are shown as a function of bins for the WVR absolute gradient in the upper graphs. The number of data points in each bin is shown in the graphs below.

gradients and VLBI data, respectively. Lu et al. (2016) reported values of the correlation coefficient between gradients from multi-GNSS and fron the European Centre for Medium-Range Weather Forecasts (ECMWF). The largest value was 0.63.
Using data obtained from only two months (May and June), Kačmařík et al. (2019) found that the GNSS gradients have correlation coefficients with the ERA5 gradients varying between 0.65 to 0.74. Elgered et al. (2019) noted that the correlation coefficients between wet gradients estimated from GPS and WVR data can reach up to 0.8 in certain months. They observed a strong seasonal dependence, with coefficients ranging from 0.3 during months with smaller gradients to 0.8 in the warmer, more humid months of the year. Teke et al. (2013) identified distinct differences in the correlation between GNSS and WVR
gradients across different stations which they attributed in part to variations in humidity and its temporal and spatial variability. This pattern is consistent with our results in Fig. 7. Both correlation and SD increase when the gradients are large.

## 5  Conclusions

The comparison between gradients derived from GNSS stations and WVR observations reveals several key points. The strong agreement among GNSS stations indicates the robustness of GNSS-based gradient estimation with no significant impacts from

the different antenna installations of the stations. When comparing GNSS gradients with WVR gradients under varying LWC thresholds, it shows that lower LWC thresholds enhance the quality of WVR data, but with the cost of reduced data availability.

The impact of applying different constraints in GNSS data processing is significant. A strong constraint reduces random errors but may not capture rapid atmospheric changes effectively. Conversely, a weak constraint allows for better tracking of short-lived gradients but introduces higher formal errors. This may be desirable for weather nowcasting applications, where rapid updates on atmospheric moisture are crucial for short-term weather predictions and which often require highly responsive data on water vapor variability, even if there is a trade-off with formal errors (Guerova et al., 2016).

The result also shows that the averaged GNSS gradients from eight co-located stations can significantly reduce the random noise introduced by a weak constraint and outperforms the undifferentiated gradients from a single station in terms of correlation and standard deviation when compared to the WVR data. This fusion approach can be used in order to obtain a more reliable variability of the water vapour for a GNSS station which can be applied as more suitable constraint for future GNSS data processing. Such an approach can also be applied to a single GNSS station with multiple receivers connected to the same antenna (Wang et al., 2024). If the random noise is mainly caused by receivers, the antenna environment, or something else, remains to be investigated.

*Acknowledgement.* We acknowledge the valuable input from the reviewers. Several issues in Section 4 are discussed based on their questions and comments. The geodetic research infrastructure at the Onsala Space Observatory is funded by Swedish Mapping, Cadastral and Land Registration Authority and Chalmers University of Technology.

*Data availability.* The estimated gradients from the GNSS and the WVR data are archived by the Swedish National Data Service (Elgered and Ning, 2025)

*Author contributions.* The two authors (TN and GE) planned the work and the structure of the paper together. TN performed the GNSS data analyses and GE performed the WVR data analyses. Both contributed to the writing of the manuscript and approved it before the submission.

*Competing interests.* The authors declare that they have no conflict of interest.

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
