# Peer review of "Atmospheric horizontal gradients measured with eight co-located GNSS stations and a microwave radiometer"

_EGUsphere, 2024_

## Referee Comment (RC1)

**EGU Sphere 2024-2716 Paper – Review**

4 November 2024

**Summary of the Study**

The paper titled *"Atmospheric horizontal gradients measured with eight co-located GNSS stations and a microwave radiometer"* by Tong Ning and Gunnar Elgered presents an investigation of atmospheric horizontal gradients measured by eight co-located GNSS (Global Navigation Satellite System) stations and a microwave radiometer (WVR) at the Onsala Space Observatory in Sweden.

The study includes a comprehensive dataset composed of eight GNSS stations and a WVR, all co-located, and covers a 22-month period. This provides a robust dataset for evaluating atmospheric gradients, including all typical weather conditions that may happen during a year and observed across different GNSS station designs and processing configurations.

The authors investigate the agreement between GNSS-derived atmospheric gradients and those obtained from the WVR, the impact of the setup of constraints on GNSS tropospheric gradient estimation, and the added value of fusing gradients obtained from the 8 GNSS stations to improve correlation with the WVR results by reducing random errors.

The authors also nuanced their finding considering the impact that the Liquid Water Content (LWC) can have on the WVR-based gradient retrieval accuracy and mentioned the WVR data cleaning after a rainy period (rain drop drying), which affect the WVR capability to retrieve accurate Zenith Wet Delay (ZWD) and horizontal gradients, hence the inter-comparison.

The methodology used for the paper is usually quite clearly stated but the paper somehow misses to identify/mention clearly the overall goal(s) targeted by the study (beyond inter-comparing horizontal gradients of two techniques, what do the authors expect to demonstrate? For which application(s)? How results can be useful/(re)used by potential readers? Etc.) and the novelties the authors bring with respect to previous studies or existing literature.

The discussion and interpretation parts are also somehow limited. As an example, the seasonal break down of the results via monthly means is a good start but the authors somehow stopped half-way without valorising their presented results: Indeed, one naturally expect that with (physical) gradients of larger magnitude the correlation between techniques will also rise (i.e., increasing the S/N will reduce the (relative) impact of random errors on the inter-comparisons) but a reader would certainly like to read more on this. E.g., at which threshold of amplitude an estimated gradient will be dominated by a real physical signal/information content, hence if the level of random errors can be "neglected" or if a user can cope with it?

The authors may also have elaborated more on the impact of the two different observation schemes of the WVR (see Figure 3) and the impact of the scheme differences on the gradient inter-comparison. Does it explain part of the bias? Do the authors see no difference because of the time averaging (monthly means)? It hard to believe that on short timespan (let's say 5 to 15

min) there is no impact of the scheme differences on the gradient inter-comparison, hence an impact on some possible applications (e.g., authors mention NWP data assimilation in their conclusion). Unless, the GNSS processing also applies a quite high cut-off angle.

Also, the monthly aggregation underlines the hypothesis that the amplitude of gradients is seasonally dependent, but it is not supported by any literature reference in the paper. While the amplitude of the gradients might have seasonality's, their amplitudes are surely also largely influenced on a weather-event based. Hence, it might have been better to aggregate the results, not by month, but in different bins of the amplitude of the gradients, which would have been physically closer to some classification such as stable/.../moderate/.../severe gradient conditions, independently if these events happen in a certain month or another one. This would enhance and strengthen the author's findings.

Concerning the literature, the paper would also be strengthened if the values obtained by the authors could be compared with other manuscripts. How do these inter-comparison values compare? The use of ERA5 data to estimate the hourly gradients could also add value to the manuscript with a 3rd independent dataset.

To conclude, the paper makes an interesting contribution to understanding atmospheric gradients by examining GNSS and WVR tropospheric gradient datasets. The methodology is sound, and the findings quite straightforward. Addressing the limitations mentioned above would further strengthen the study's implications and relevance and would enhance clarity and interpretability. It is recommended to address these limitations prior to publication. Below, authors will find some additional specific remarks.

**Specific remarks:**

- Introduction:
    - The sentence "The investigation demonstrates [...] general data assimilation enhancements" is not clear. What do you mean? That assimilating also gradients in NWP brings an added value? Rewording would bring clarity.
    - Some additional references can surely be added into the list at line 29. E.g., the COST Action GNSS4SWEC final dissemination report can be used for summarising several of them.
    - Line 29: the "the satellite geometries of GNSS measurements [...]" expression is not clear/correct and can be reworded with e.g., something like "the remote sensing of the atmosphere at a given station is improved as more GNSS constellations is added, by increasing the number of simultaneous measurements and their spreading in all directions, hence benefiting to the gradient estimation".
    - Line 31: "Glonass" is usually written "GLONASS" in the literature.
- Datasets section:
    - Line 66: "The input to the data processing" is not correct. The input is raw GNSS observation. You can probably reword with something like "The GNSS data processing uses ionospheric free linear combinations".

- o Line 67: "atmospheric parameters". Be more specific (ZTD+gradients).
- o Line 71: "Equal weighting of the observations was applied". Do you mean that no elevation dependent weighting is applied during the data processing?
- o Line 82: You mention that the VMF data server (2024) is using ERA-Interim. Isn't it not ERA5? The ECMWF web site mention that "ERA-Interim production stopped on 31st August 2019". If it is ERA5, then you must change ERA-Interim everywhere necessary in the text.
- o Line 90: You mention that part of the WVR time series has been recorded with an elevation angle above 25°. Won't you expect that it has an influence on the GNSS-WVR gradient bias (unless the cut-off angle in the GNSS data processing is also high, but usually we would use 3°)?
- o Line 110: Not clear to me what you mean by "synchronisation of the datasets".
- o Line 111: "In addition the pure GNSS datasets" → "In addition to the GNSS-based datasets, we use".
- Comparison section:
  - o Lines 135-136: "The rms differences are […] standard deviations." You state it but you don't interpret. What is your actual finding(s) with that?
  - o Title of section 3.2: you can remove the part "for LWC […] 0.7 mm" for readability.
  - o Line 147: missing the symbol "<" before "0.05" and "0.7 mm".
  - o Lines 152-153: "The explanation […] of 0.1°". This is not clear to me: do you mean that the recorded elevation angle of a given WVR data has an uncertainty of 0.1° compared to the actual elevation angle? If yes, do you really think it can explain the bias between WVR and GNSS? This is also related to my previous point mentioning that part of the WVR data was recorded with an elevation angle set above 25°. I think this is worth some discussion in the paper.
  - o Line 166: "what does "the total gradient from the WVR" means? Doesn't the WVR provide only the wet gradients? And don't you remove the hydrostatic gradients from the GNSS total gradients?
  - o Lines 173 and following: the fusion procedure is not that easy to understand. It can be clearer.
  - o Figure 8: Dates below the graphs can be smaller to enhance readability. Even, they can be at 90° of the horizontal axes.
- Conclusion:
  - o Line 203: "for applications with higher […] errors". Which applications?
  - o Lines 204-206: I can understand that you tried to reference to the Product Requirement Document (PDR) of E-GVAP but it doesn't include requirements for gradients, only for ZTD/ZWD/IWV. The mentioned 15mm threshold is for ZTD and your paper focus only on gradients.
- Everywhere in the text: Don't use the expression "GNSS data" or "GNSS measurements" when you refer to e.g., gradients as it is confusing with raw GNSS observations. Please use the exact wording instead. Similar remark can be done with e.g., WVR data.
- How acronyms are cited can be harmonized. E.g., the text mention "GIA (glacial isostatic adjustment)" and "Zenith Total Delays (ZTD)". Please, choice one way of citing acronyms

and ensure that the first letter of the words is capitalized or not everywhere (according to journal's rule).

- Kierulf et al. Is mentioned in the text with the date 2019 while in the reference list it is 2021.

---

## Author Comment (AC1)

Dear Editor / Reviewers,

Please find the uploaded revised version of our manuscript egusphere-2024-2716 together with a marked-up version of the manuscript is provided in order to show the changes made.

We appreciate the work of the reviewers, their questions and their suggestions for improvements. We chose to include responses to both reviewers in the same document.

In summary, both reviewers pointed out some errors and need for clarifications. They have all been adopted and further discussed as described below. Both reviewers also pointed out the need for more discussions and interpretations of the results. We have adopted the suggestion by Reviewer 2, to add a section with the title Discussion. Here we further develop our interpretations of the results, such as the specific shortcomings of the radiometer (pointing uncertainties, the elevation cutoff angle, and the restriction on the used azimuth angles) and how they affect the comparison results. The figure with monthly means of gradients was expanded and moved from Section 2 (Datasets) to Section 4 (Discussion, now Figure 8) because if its relevance to the gradient biases we believe are cause by the WVR, Two new figures (9 and 10) are also added to the discussion as a result of input from the reviewers suggesting an alternative way to present correlations and standard deviations in terms of their dependence on the size of the gradients.

Some text from the conclusions in the original manuscript was removed and the corresponding issues are handled in the discussion.

Sincerely yours

T. Ning and G. Elgered

**RESPONSES**
**Reviewer # 1:**

General comments:

The paper titled "Atmospheric horizontal gradients measured with eight co-located GNSS stations and a microwave radiometer" by Tong Ning and Gunnar Elgered presents an investigation of atmospheric horizontal gradients measured by eight co-located GNSS (Global Navigation Satellite System) stations and a microwave radiometer (WVR) at the Onsala Space Observatory in Sweden.

The study includes a comprehensive dataset composed of eight GNSS stations and a WVR, all co-located, and covers a 22-month period. This provides a robust dataset for evaluating atmospheric gradients, including all typical weather conditions that may happen during a year and observed across different GNSS station designs and processing configurations.

The authors investigate the agreement between GNSS-derived atmospheric gradients and those obtained from the WVR, the impact of the setup of constraints on GNSS tropospheric gradient estimation, and the added value of fusing gradients obtained from the 8 GNSS stations to improve correlation with the WVR results by reducing random errors.

The authors also nuanced their finding considering the impact that the Liquid Water Content (LWC) can have on the WVR-based gradient retrieval accuracy and mentioned the WVR data cleaning after a rainy period (rain drop drying), which affect the WVR capability to retrieve accurate Zenith Wet Delay (ZWD) and horizontal gradients, hence the inter-comparison.

The methodology used for the paper is usually quite clearly stated but the paper somehow misses to identify/mention clearly the overall goal(s) targeted by the study (beyond inter-comparing horizontal gradients of two techniques, what do the authors expect to demonstrate? For which application(s)? How results can be useful/(re)used by potential readers? Etc.) and the novelties the authors bring with respect to previous studies or existing literature.
*Response: In the updated manuscript we now have added a new section "Discussion" and rewritten the section "Conclusion" to address these issues. We have highlighted the use of gradients to assess the performance of different GNSS station installations and the fusion method used to decrease the relative importance of random errors. We have also added other interpretations more related to the shortcomings of the method.*

The discussion and interpretation parts are also somehow limited. As an example, the seasonal break down of the results via monthly means is a good start but the authors somehow stopped half-way without valorising their presented results: Indeed, one naturally expect that with (physical) gradients of larger magnitude the correlation between techniques will also rise (i.e., increasing the S/N will reduce the (relative) impact of random errors on the inter-comparisons) but a reader would certainly like to read more on this. E.g., at which threshold of amplitude an estimated gradient will be dominated by a real physical signal/information content, hence if the level of random errors can be "neglected" or if a user can cope with it?
*Response: In the Discussion section we have added two more figures (Figures 9 and 10) to illustrate of how the correlation depends on the amplitude of observed the gradients. The results show that the correlation coefficients between the GNSS and WVR gradients depend not only on errors of the specific techniques but also on the size and variability of the true*

*gradients. Additionally, because different volumes of air are sampled, the correlation is influenced by how well the linear model represents reality.*

The authors may also have elaborated more on the impact of the two different observation schemes of the WVR (see Figure 3) and the impact of the scheme differences on the gradient inter-comparison. Does it explain part of the bias? Do the authors see no difference because of the time averaging (monthly means)? It hard to believe that on short timespan (let's say 5 to 15 min) there is no impact of the scheme differences on the gradient inter-comparison, hence an impact on some possible applications (e.g., authors mention NWP data assimilation in their conclusion). Unless, the GNSS processing also applies a quite high cut-off angle.
*Response: We have studied the two subsets for the different observation schemes of the WVR. We see no significant impact on either the correlation or the standard deviation. However, the lower cutoff angle will make the WVR gradient estimates more sensitive to pointing errors. This is discussed in more detail now and we have added Figure 8, showing an increase in mean value of the monthly mean east gradient during the months at the end of the dataset when the cutoff elevation angle is $20^\bullet$ instead of $30^\bullet$.*

Also, the monthly aggregation underlines the hypothesis that the amplitude of gradients is seasonally dependent, but it is not supported by any literature reference in the paper. While the amplitude of the gradients might have seasonality's, their amplitudes are surely also largely influenced on a weather-event based. Hence, it might have been better to aggregate the results, not by month, but in different bins of the amplitude of the gradients, which would have been physically closer to some classification such as stable/.../moderate/.../severe gradient conditions, independently if these events happen in a certain month or another one. This would enhance and strengthen the author's findings.
*Response: The new Figures 9 and 10 have been added in the Discussion section to illustrate the relation between gradient size and correlation.*

Concerning the literature, the paper would also be strengthened if the values obtained by the authors could be compared with other manuscripts. How do these inter-comparison values compare? The use of ERA5 data to estimate the hourly gradients could also add value to the manuscript with a 3rd independent dataset.
*Response: In the study, we used the ERA5 data to confirm that the biases observed (approximately equal to the mean gradients from the WVR), mainly in the east gradient, is likely caused by the WVR. Additional comparisons between ERA5 and GNSS show larger standard deviations and lower correlations (compared to GNSS vs WVR) and do not add any additional information in terms of distinguishing the quality of the eight GNSS stations or affecting the results of the fusion of the eight stations. Although it is not the focus of our study it does not mean that it is not of interest from an ERA5 perspective, and the data are open access, so anyone interested have the possibility to perform such comparisons.*

To conclude, the paper makes an interesting contribution to understanding atmospheric gradients by examining GNSS and WVR tropospheric gradient datasets. The methodology is sound, and the findings quite straightforward. Addressing the limitations mentioned above would further strengthen the study's implications and relevance and would enhance clarity and interpretability. It is recommended to address these limitations prior to publication. Below, authors will find some additional specific remarks.

Specific comments:

Introduction:

The sentence "The investigation demonstrates [...] general data assimilation enhancements" is not clear. What do you mean? That assimilating also gradients in NWP brings an added value? Rewording would bring clarity.

*Response: The sentence has been rephrased to "The investigation demonstrates that assimilation of GNSS ZTD can benefit from enhancements in general data assimilation techniques. This can lead to improved forecast quality and more accurate numerical weather predictions (NWP)".*

Some additional references can surely be added into the list at line 29. E.g., the COST Action GNSS4SWEC final dissemination report can be used for summarising several of them.

*Response: A new reference "COST Action GNSS4SWEC: Advanced GNSS Tropospheric Products for Monitoring Severe Weather Events and Climate; editing status 2024-05-15; Springer; https://link.springer.com/book/10.1007/978-3-030-13901-8" has been added.*

Line 29: the "the satellite geometries of GNSS measurements [...]" expression is not clear/correct and can be reworded with e.g., something like "the remote sensing of the atmosphere at a given station is improved as more GNSS constellations is added, by increasing the number of simultaneous measurements and their spreading in all directions, hence benefiting to the gradient estimation".

*Response: The sentence has been rewritten to "In addition, the remote sensing of the atmosphere at a given station improves with the addition of more GNSS constellations. This enhancement comes from increasing the number of simultaneous measurements and their distribution in various directions, which benefits the gradient estimation."*

Line 31: "Glonass" is usually written "GLONASS" in the literature.
*Response: Yes, we now write GLONASS.*

Datasets section:

Line 66: "The input to the data processing" is not correct. The input is raw GNSS observation. You can probably reword with something like "The GNSS data processing uses ionospheric free linear combinations".

*Response: The sentence has been rephrased to "The processing uses ionospheric free linear combinations formed by acquired GNSS phase-delay observations while the output included station coordinates, clock biases, and atmospheric parameters, i.e., ZTD and linear horizontal gradients."*

Line 67: "atmospheric parameters". Be more specific (ZTD+gradients).
*Response: See response above.*

Line 71: "Equal weighting of the observations was applied". Do you mean that no elevation dependent weighting is applied during the data processing?

*Response: Yes! It is now clarified in the text. We did not apply any elevation-dependent weighting of the GPS observations based on the conclusion given by Elgered et al. (2019) where they found that the GNSS solution without weighting gives a better agreement with the WVR gradients compared to the solution with elevation-dependent weighting.*

Line 82: You mention that the VMF data server (2024) is using ERA-Interim. Isn't it not ERA5? The ECMWF web site mention that "ERA-Interim production stopped on 31st August 2019". If it is ERA5, then you must change ERA-Interim everywhere necessary in the text.
*Response: The web page https://vmf.geo.tuwien.ac.at/products.html states ERA-Interim, which however probably needs to be updated. It is correct that the ERA-interim stopped in August 2019 https://www.ecmwf.int/en/newsletter/159/meteorology/global-reanalysis-goodbye-era-interim-hello-era5. We changed to ERA5 in the text.*

Line 90: You mention that part of the WVR time series has been recorded with an elevation angle above 25°. Won't you expect that it has an influence on the GNSSWVR gradient bias (unless the cut-off angle in the GNSS data processing is also high, but usually we would use 3°)?
*Response: This was an error, or at least unclear. We now discuss the two observing schemes for the WVR in more detail, both in Section 2 (Datasets) and Section 4 (Discussion) because the elevation cutoff angle affects how a pointing error will propagate into the gradient estimate.*

Line 110: Not clear to me what you mean by "synchronisation of the datasets".
*Response: The sentence has been rephrased to "Horizontal gradients can change rapidly and there are data gaps. Therefore, it is important to synchronize the datasets from the stations being compared to ensure that the results obtained only use data points when observations are available from all stations."*

Line 111: "In addition the pure GNSS datasets" → "In addition to the GNSS-based datasets, we use".
*Response: Changed*

Comparison section:
Lines 135-136: "The rms differences are [...] standard deviations." You state it but you don't interpret. What is your actual finding(s) with that?
*Response: We have modified the text to "meaning that the biases are small, of the order of 0.1 mm or less, which indicates that there is no permanent severe multipath effects in a specific direction at any of the eight stations."*

Title of section 3.2: you can remove the part "for LWC [...] 0.7 mm" for readability.
*Response: Done*

Line 147: missing the symbol "<" before "0.05" and "0.7 mm".
*Response: Done*

Lines 152-153: "The explanation [...] of 0.1°". This is not clear to me: do you mean that the recorded elevation angle of a given WVR data has an uncertainty of 0.1° compared to the actual elevation angle? If yes, do you really think it can explain the bias between WVR and GNSS? This is also related to my previous point mentioning that part of the WVR data was recorded with an elevation angle set above 25°. I think this is worth some discussion in the paper.
*Response: Yes, even a small pointing offset will affect the gradient because the gradients are small compared to the ZWD. We have added text in the Discussion section on this. In*

*short, the sensitivity to pointing offsets increase with a decreasing elevation cutoff angle and especially for east-west gradients because of the restriction to observe at azimuth angles between 0 and 180 degrees. (The position of the GNSS satellites on the sky is known with a superior accuracy compared to the WVR.) We specifically state that "... pointing errors of the WVR are a major source of error when WVR gradients are compared to those from GNSS in an absolute sense".*

Line 166: "what does "the total gradient from the WVR" means? Doesn't the WVR provide only the wet gradients? And don't you remove the hydrostatic gradients from the GNSS total gradients?

*Response: We were referring to the total wet gradient amplitude, but while updating the manuscript we have realized that it will be more correct to present the comparisons based on different subsets of the data when the WVR gradient in the east and the north directions are larger than a specific value. This is because there may very well be a large gradient in one of the directions while it is insignificant in the other direction and shall therefore not be included in that subset of the data. The result of this change is that the total wet gradient from the WVR is no longer used in the manuscript.*

Lines 173 and following: the fusion procedure is not that easy to understand. It can be clearer.

*Response: The sentences have been expanded to explain the fusion procedure as: The procedure starts with comparison between WVR gradients and gradients from a single station (ONSA). Thereafter, we took the gradients from the ONSA station and calculated an average with the ones from the ONS1 station. This new set of average gradients was then compared to the WVR gradients to assess their agreement. Then we incorporated the gradients from the OTT1 station into the previously averaged data from ONSA and ONS1, creating a new combined average and compared to WVR gradients again. We continued this iterative process by gradually adding the gradients from additional stations, one by one, each time recalculating the average and comparing it with the WVR gradients. This process was repeated until we had included the gradients from all eight GNSS stations. "*

Figure 8: Dates below the graphs can be smaller to enhance readability. Even, they can be at 90° of the horizontal axes.

*Response: Figure 8 (now Figure 7) has been modified with a smaller size on dates which are also at 90° of the horizontal axes.*

Conclusion:
Line 203: "for applications with higher [...] errors". Which applications?

*Response: The sentence has been rewritten to "This may be desirable for weather nowcasting applications, where rapid updates on atmospheric moisture are crucial for short-term weather predictions and which often require highly responsive data on water vapor variability, even if there is a trade-off with formal errors (Guerova et al., 2016)."*

Lines 204-206: I can understand that you tried to reference to the Product Requirement Document (PDR) of E-GVAP but it doesn't include requirements for gradients, only for ZTD/ZWD/IWV. The mentioned 15mm threshold is for ZTD and your paper focus only on gradients.

*Response:  The sentence has been removed.*

Everywhere in the text: Don't use the expression "GNSS data" or "GNSS measurements" when you refer to e.g., gradients as it is confusing with raw GNSS observations. Please use the exact wording instead. Similar remark can be done with e.g., WVR data.
***Response: We have changed to GNSS gradients and WVR gradients where applicable.***

How acronyms are cited can be harmonized. E.g., the text mention "GIA (glacial isostatic adjustment)" and "Zenith Total Delays (ZTD)". Please, choice one way of citing acronyms and ensure that the first letter of the words is capitalized or not everywhere (according to journal's rule).
***Response: Done***

Kierulf et al. Is mentioned in the text with the date 2019 while in the reference list it is 2021.
***Response: 2021 is the correct year. However, we decided to remove this citation when we added the reference to the Springer Handbook with a lot of different GNSS applications. Although the BIFROST project is an important application for the SWEPOS network, glacial isostatic adjustment (land uplift) is not of specific interest to this study.***

**Reviewer # 2:**

General comments:

I want to thank the authors for an interesting manuscript. I agree the topic is worth of investigation and publication. The study is well performed and the manuscript mostly well written. Still, I would like to ask authors to address some comments before the publication.

Specific comments:

L17: authors mention various applications of GNSS data usage. It would be worthy for some readers to speficically name at least some of the applications.***Response: The sentence has been modified to "After decades of development, data acquired from continuously operating Global Navigation Satellite System (GNSS) stations are widely used in various applications, including precise positioning for navigation, real-time tracking for transportation logistics, environmental monitoring, including climate studies, geodesy, and geophysical research to understand the dynamics of the Earth's crust (Teunissen and Montenbruck, 2017)."***

L18: "GPS receivers" are mentioned - is it really only GPS receivers? Would it not be possible to use the term GNSS receiver here?
***Response: Yes, corrected.***

L22: the manuscript is focusing on GNSS derived horizontal tropospheric gradients. Their estimation and utilizitaion has been the subject of scientific studies in the last ten years or so. It would therefore be useful to extend the Introduction section in this regard and provide a better prepared summary of the current state of the art in this area (on top of several already cited works).
***Response: The introduction has been extended focusing on the GNSS derived horizontal tropospheric gradients. "Furthermore, the usage of GNSS data to estimate horizontal tropospheric gradients has become increasingly prevalent in recent years. These gradients, which represent the asymmetry of signal delays in the azimuth direction contains information on the local meteorological conditions, and are crucial for improving the accuracy of GNSS-derived ZTD. Studies have shown that the incorporation of GNSS tropospheric gradients can enhance meteorological applications, such as weather forecasting and climate research."***

L65: I miss some information about the GNSS data processing:
- definitely you should provide elevation cut-off angle as it can impact estimation of gradients
- what about the interval of input observations? Was it 30s or different?
- did you process 24h data or different time periods? If 24h, have you somehow dealt with the problem of decreased quality of tropospheric parameters estimated at the boundary of the day?
- what about PCV/PCO corrections?
***Response: All missing information is added to the updated manuscript. We are using 24 h data for each day and based on comparison with WVR data and we accept the slightly lower quality after inspection of the time series. The main point is that we treat all the 8 GNSS stations in the same way.***

L71: what exactly do you mean with sentence "Equal weighting of the observations was applied"? Do you mean observations from individual GNSS systems? And/or that you applied no elevation dependent weighting?

*Response: We did not apply any elevation-dependent weighting to the GPS observations based on the conclusion given by Elgered et al., 2019 where they found that the GNSS solution without weighting gives a better agreement with the WVR gradients compared to the solution with elevation-dependent weighting. This clarification has now been included in the updated manuscript.*

L81: you mention usage of hydrostatic tropospheric gradients from VMF data server. Please, consider adding some information on what these gradients looked like, for example in the form of basic statistics (min, max, mean, sdev).

*Response: One sentence is added to give some statistics of hydrostatic gradients. "During the study period, we found that the hydrostatic east gradients ranged from −1.1 mm to 0.7 mm, with a mean of −0.1 mm and an SD of 0.3 mm. Similarly, the hydrostatic north gradients ranged from −1.2 mm to 0.5 mm, with a mean of −0.3 mm and an SD of 0.3 mm."*

L91: I wonder why the elevation cut-off for WVR measurements was 25 degrees. Please, can you explain in the manuscript? I guess that for GNSS data processing you applied a much lower elevation cut-off (probably something between 3 and 10 degrees). Can't the different cut off angle cause part of the difference between GNSS and WVR results? Please, discuss in the manuscript.

*Response: This was an error, "above 25 degrees" was fact 30 degrees during the first 18 months. Thereafter the lowest elevation was 20 degrees, The description of the WVR pointing is updated and more detailed.*

Figure 3: on the right figure it seems to me that on azimuth of 45°, the measurement on the lowest elevation is missing. What is the reason? Some blockage of the sky view around the WVR?

*Response: Yes. The missing observation at the lowest elevation angle at the azimuth angel 45° is due to the blockage of a radio telescope. This information now is added to the caption of the figure.*

L151: why do you expect "the true values of the mean gradients to be close to zero over a time period of almost two years"? What is your reasoning behind this statement? I would expect that mean wet gradients would point to the south on the northern hemisphere (and north on the southern hemisphere) due to global distribution of water vapour (higher values around the equator compared with higher latitudes).

*Response: We expect that the mean values of horizontal gradients at a GPS station to be close to zero. This is because the gradients, which indicate the asymmetry in water vapour distribution, tend to fluctuate around a mean value of zero over long time periods due to the balanced variations in atmospheric conditions. Given that most of the water vapour is found in the first few kilometres a north-south gradient caused by higher temperatures towards the south is insignificant. Probably the coastline location is more important. The average winter and summer gradients may have opposite signs since the air above the sea is warmer in the winter compared to the land and colder in the summer. A new plot has been added in the updated manuscript to show mean values from GNSS fusion gradients with 8 stations for each month for both east and north gradients. Although it is not a clear "detection" the results indicate that this may be the case.*

Table 3: you present RMS differences for gradients from individual GNSS stations. It can be hard for the reader to interpret them without knowing the typical range of values of the gradient themselves. I therefore suggest to provide information about the standard values of Ge and Gn over the studied period. It can be just basic statistics for Gn/Ge for the single GNSS station (e.g. ONSA). I know you provide such an information by Figure 5, but for the so called gradient amplitudes, not Ge/Gn values themselves. Do you have any explanation for Ge (east gradient) having a slightly better aggrement than Gn (north gradient)?

*Response: The basic statistics for Gn/Ge are added for ONSA as an example. " During the study period, the mean value of the east gradients, for the ONSA station, is −0.01 mm with an SD of 0.54 mm, while the north gradients have a mean of −0.06 mm with an SD of 0.50 mm." We see a slightly better agreement for the east gradients, possibly because of the poorer sampling of the sky north of the zenith direction due to the geometry of the GNSS satellite constellation at this latitude. This statement is now added to the updated manuscript with references.*

Table 4: correlation coefficients for GNSS vs. WVR comparison are not that high (typically 0.6 - 0.7). Although you compare the correlation coefficients between the two constraining options, LWC values, etc. In your writing, you do not specifically mention or comment the values themselves. Please, consider it.

*Response: A new discussion section has been added in the updated manuscript to make comments on the resulting correlation coefficients. We also make comparisons with the findings from other studies. The bottom line is that correlations coefficients can be very high if the dataset include weather conditions with large gradients only. For a large dataset like ours there will also be a lot of observations when gradients are small or insignificant.*

L189: I find the manuscript lacking a deeper discussion, which is inevitable in any scientific writing. Please include it in a separate added section named Discussion, or expand the Conclusion section with it. Some of the topics which can be discussed are mentioned in my comments above.

*Response: In the updated manuscript we now have added a new section "Discussion" and rewritten the section "Conclusion" to address several of the issues from both reviewers. We have highlighted the use of gradients to assess the performance of different GNSS station installations and the fusion method used to decrease the relative importance of random errors in the Conclusions. We have also added other interpretations more related to the shortcomings of the method in the Discussion.*